# Association of Allostatic Load with All-Cause and Cancer Mortality by Race and Body Mass Index in the REGARDS Cohort

**DOI:** 10.3390/cancers12061695

**Published:** 2020-06-26

**Authors:** Tomi Akinyemiju, Lauren E Wilson, April Deveaux, Stella Aslibekyan, Mary Cushman, Susan Gilchrist, Monika Safford, Suzanne Judd, Virginia Howard

**Affiliations:** 1Department of Population Health Sciences, Duke University School of Medicine, Durham, NC 27701, USA; 2Duke Cancer Institute, Duke University, Durham, NC 27710, USA; lauren.e.wilson@duke.edu (L.E.W.); april.deveaux@duke.edu (A.D.); 3Department of Epidemiology, University of Alabama at Birmingham, Birmingham, AL 35233, USA; saslibek@uab.edu (S.A.); vjhoward@uab.edu (V.H.); 4Department of Medicine and University of Vermont Cancer Center, Larner College of Medicine at the University of Vermont, Burlington, VT 05405, USA; Mary.Cushman@uvm.edu; 5Department of Clinical Cancer Prevention and Cardiology, University of Texas MD Anderson Cancer Center, Houston, TX 77030; USA; sgilchrist@mdanderson.org; 6Weill Cornell Medical College, Weill Cornell, New York City, NY 10065, USA; mms9024@med.cornell.edu; 7Department of Biostatistics, University of Alabama at Birmingham, Birmingham, AL 35233, USA; sejudd@uab.edu

**Keywords:** allostatic load, mortality, racial disparities, obesity, biomarkers

## Abstract

Among 29,701 Black and White participants aged 45 years and older in the Reasons for Geographic and Racial Difference in Stroke (REGARDS) study, allostatic load (AL) was defined as the sum score of established baseline risk-associated biomarkers for which participants exceeded a set cutoff point. Cox proportional hazard regression was utilized to determine the association of AL score with all-cause and cancer-specific mortality, with analyses stratified by body-mass index, age group, and race. At baseline, Blacks had a higher AL score compared with Whites (Black mean AL score: 2.42, SD: 1.50; White mean AL score: 1.99, SD: 1.39; *p* < 0.001). Over the follow-up period, there were 4622 all-cause and 1237 cancer-specific deaths observed. Every unit increase in baseline AL score was associated with a 24% higher risk of all-cause (HR: 1.24, 95% CI: 1.22, 1.27) and a 7% higher risk of cancer-specific mortality (HR: 1.07, 95% CI: 1.03, 1.12). The association of AL with overall- and cancer-specific mortality was similar among Blacks and Whites and across age-groups, however the risk of cancer-specific mortality was higher among normal BMI than overweight or obese participants. In conclusion, a higher baseline AL score was associated with increased risk of all-cause and cancer-specific mortality among both Black and White participants. Targeted interventions to patient groups with higher AL scores, regardless of race, may be beneficial as a strategy to reduce all-cause and cancer-specific mortality.

## 1. Introduction

Allostatic load (AL) describes the physiological burden of cumulative stress on biological systems normally involved in adaptation to environmental challenges [1]. Under normal circumstances, human physiological systems adapt to environmental challenges, such as diurnal rhythms, environmental toxins, psychosocial stress, etc. However, it is well documented that accumulated stress may contribute to the breakdown of the normal physiological adjustments—a term known as allostasis—leading to physiological dysregulation across several biological systems, including immune, cardiovascular, and metabolic [1,2]. The dysregulation of these systems influences tumor-promoting inflammation, immune regulation and cellular energetics, which have been well described as hallmarks of cancer [3]. The measurement of allostatic load has been conceptualized as an increasing number of biomarkers, indicating glucocorticoid dysregulation including elevated blood pressure, pulse rate, total cholesterol levels, C-reactive protein, serum creatinine, and blood urea nitrogen, among others [1,2,4,5]. Epidemiologic studies have also shown that inflammatory [6,7] and metabolic dysregulation [8,9], components of AL, increase the risk of all-cause and cancer-specific mortality. However, there are considerable gaps in our understanding of the association between AL and mortality outcomes. 

To our knowledge, only seven studies have directly evaluated the association between AL and mortality outcomes [5,10,11,12,13,14,15], five of which focused on US populations. Three of those five studies utilized NHANES data [10,12,14], two utilized data from a cohort study of older Americans [5,13], while two non-US studies included populations in the United Kingdom [11,15]. Across the seven studies, higher AL was consistently associated with increased risk of all-cause or cause-specific mortality. Only one US study has directly evaluated racial differences in the association between AL and mortality outcomes. Using NHANES III data, this study observed that higher AL was associated with an increased risk of all-cause mortality among Whites but not Blacks [10]. Body mass index (BMI) is another well-documented risk factor for mortality, including cancer mortality, and contributes to chronic inflammation and other metabolic alterations that constitute component measures of AL [8,16]. Racial disparities have also been well documented in obesity rates among US adults [17]. While the burden of AL is higher among Blacks compared with Whites [18], the association between AL and all-cause or cancer-specific mortality, and potential effect modification by race/ethnicity and BMI remains understudied. In this study, we examined the association of baseline AL with all-cause and cancer-specific mortality in a nationwide prospective cohort and assessed whether the associations varied by BMI or race/ethnicity.

## 2. Results

Among 29,701 participants (mean follow-up: 6.5 years, SD: 2.2), 4622 deaths occurred and 1237 were due to cancer. At baseline, Black participants had higher mean AL than White participants, overall (Black mean AL score: 2.42, SD: 1.50; White mean AL score: 1.99, SD: 1.39), across age and BMI categories, as shown in Figure 1, and across most demographic and risk factor characteristics defined (*p* < 0.01), as shown in Table 1. Those with high AL (score ≥ 3) were more likely to be lower income, sedentary, current or past smokers, have diabetes, and be overweight or obese, as shown in Table 2. They were also less likely to consume alcohol and to have completed high school. In fully adjusted models, shown in Table 3, every unit increase in AL score was associated with a 24% higher risk of all-cause mortality (HR: 1.24, 95% CI: 1.22, 1.27) and 7% higher risk of cancer-specific mortality (HR: 1.07, 95% CI: 1.03, 1.12). When stratified by race, increasing AL score was associated with a 26% increased risk of all-cause mortality among Black participants (HR: 1.26, 95% CI 1.22, 1.30) and a 23% increased risk among White participants (HR: 1.23, 95% CI 1.20, 1.27). Every unit increase in AL score was associated with a 6% increased risk of cancer mortality among Black participants (HR: 1.06, 95% CI 0.99, 1.13), and a 15% increased risk among White participants (HR: 1.08, 95% CI 1.03, 1.14), as shown in Table 3. There was no evidence of effect modification by race on the association between allostatic load and mortality (data not shown); however, there was evidence of minor effect modification by BMI and age-group. 

When stratified by BMI category, shown in Table 4, every unit increase in AL score was associated with a 29% increased risk of all-cause mortality (HR: 1.29, 95% CI 1.23, 1.35) among participants with normal BMI and a 29% increased risk (HR: 1.29, 95% CI 1.26, 1.32) among participants who were overweight or obese. For cancer mortality, every unit increase in AL score increased the risk by 17% among those with normal BMI (HR: 1.17 95% CI 1.08, 1.28) and by 9% among those who were overweight/obese (HR: 1.09, 95% CI 1.03, 1.15). When stratified by age, every increase in AL score was associated with 34% higher risk of all-cause mortality among participants < 65 years (HR: 1.34, 95% CI 1.28, 1.39) and a 22% higher risk (HR: 1.22, 95% CI 1.19, 1.25) among those ≥65 years, as shown in Table 5. While magnitudes of the effect of AL score on mortality differed slightly across BMI and age strata, increasing AL score was consistently associated with increased risk of mortality for all examined patient groups. Of the 10 AL components evaluated, CRP (high vs. low HR: 1.50 95% CI 1.41, 1.59), BUN (high vs. low HR: 1.46 95% CI 1.35, 1.57), serum creatinine (high vs. low HR: 2.10, 95% CI 1.93, 2.28) and SBP (high vs. low HR: 1.34 95% CI 1.22, 1.47) were the AL component factors most consistently associated with increased risks of all-cause mortality in the total sample and among Blacks and Whites separately, while CRP (high vs. low HR: 1.30 (1.16, 1.46) was associated with increased risks for cancer mortality, as shown in Table 6.

## 3. Discussion

In the nationwide prospective REGARDS cohort, Black participants had significantly higher AL scores at baseline than White participants, and while the mean baseline AL score increased with age and BMI, it was consistently higher among Blacks compared to Whites regardless of age-group or BMI. A higher AL score was associated with a significantly increased risk of all-cause and cancer mortality after adjusting for socio-demographic and socio-economic factors. This association was observed for all-cause mortality across BMI and age categories, and for cancer mortality, among normal and overweight/obese, as well as among participants 65 years and older. Notably, the association between the composite AL score and mortality outcomes was stronger and more consistent than the association of each AL component considered separately. 

To our knowledge, this is the first prospective cohort study in the US to evaluate the association between AL and all-cause and cancer mortality among Blacks and Whites. These findings are largely in line with previous studies evaluating the associations between AL and mortality. A study using data from the MacArthur Successful Aging study observed that AL was associated with increased overall mortality in older male and female populations [5], and increased risk of overall mortality by up to 88%. A reduction in AL over a period of three years was observed to reduce the risk of overall mortality in older US adults when compared with those experiencing an increase in AL [13]. Findings from another study using NHANES III data showed stronger associations with overall mortality among Whites compared with Blacks [10], in contrast to our observation of slightly stronger point estimates for associations among Blacks compared with Whites, and no evidence of effect modification by race on the association between race and mortality. Furthermore, in contrast to the previous study, we observed significantly increased risk of all-cause mortality in both the younger and older age-groups evaluated, with a stronger association among participants younger than 65 years. We also observed significant associations between AL and cancer mortality among Blacks and Whites, a previously unreported association in a US prospective cohort. Only two other studies have previously reported on the association between AL and cancer mortality [14,15], the NHANES sample and the Scottish cohort study, with results consistent with the present study. 

The biological embodiment of chronic stress, due to racial, socio-economic, and/or psychosocial disadvantage, may significantly impact overall and cancer-specific mortality, especially among Blacks. Several studies have described a higher burden of AL as the result of repeated, chronic stresses that over-activate the sympathetic nervous system and the hypothalamus–pituitary–adrenal axis [2,19,20], leading to the dysregulation of biological systems. Predictors of high AL include race/ethnicity, adverse childhood experiences, and low SES, partially mediated through educational attainment and financial wellbeing. Persistent poverty leads to chronic stress and the disruption of the endocrine and nervous systems [21,22]. The biological mechanism underlying the association between AL and mortality likely involves alterations in the immune, metabolic, and cardiovascular systems, changes that may influence the efficiency of stress-related cardiovascular regulation [23], tumor immune-surveillance [24,25,26,27], or operate via stress-related DNA damage [28]. The association between lower SES and chronic stress with poor cancer outcomes has been previously described. For instance: Freeman et al. observed that census tract level SES explained the disparity in prostate cancer-specific survival [29]; Kelly-Irving et al. found that adverse childhood experiences were associated with cancer risk among UK women [30]; a recent study observed that symptoms of depression and cortisol levels were associated with decreased survival in metastasized renal cell carcinoma [31]. Similarly, results from a large Danish cohort showed significant associations between perceived stress levels and risk of cause-specific mortality [32]. Taken together, these findings highlight the contribution of chronic stressors to AL, ultimately resulting in increased cancer-specific mortality.

There are several strengths and limitations relevant to the interpretation of the study results. First, the prospective cohort design with a relatively long follow-up period (maximum 10 years, mean 6.5 years) makes it one of the few prospective studies to date on this topic, and the racial diversity of the cohort enabled the assessment of racial differences. A limitation of this study is that the REGARDS cohort did not include hip circumference measures, therefore we were unable to assess waist-to-hip ratio, one of the components of the AL score. We relied on waist circumference measures as a proxy in this analysis. Waist-to-hip ratio (WHR) provides a measure of visceral fat, thought to better account for differences in body structure compared with BMI, and highly correlated with chronic stress [33,34]. However, other studies suggest that WHR and waist circumference both perform better at predicting mortality outcomes compared with BMI [35,36,37]. Secondly, while AL was characterized only once at baseline, it is likely that temporal changes in AL score may also influence mortality outcomes. Future studies are needed to assess temporal changes in AL in relation to mortality outcomes among Blacks and Whites. 

## 4. Materials and Methods 

### 4.1. Study Subjects

Data for this study were obtained from the Reasons for Geographic and Racial Differences in Stroke (REGARDS) study. In 2003–2007, the REGARDS study enrolled 30,239 Black and White US adults residing in the contiguous US states, with the oversampling of Blacks and residents of the Stroke Belt and Stroke Buckle areas of the US with higher than average stroke mortality rates. The Stroke Belt is comprised of Alabama, Arkansas, Georgia, Kentucky, Louisiana, Mississippi, North Carolina, South Carolina, and Tennessee. The Stroke Buckle is a 153-county region in the coastal plains of North Carolina, South Carolina, and Georgia, with particularly elevated stroke mortality rates. The REGARDS study design and participants have been described in detail elsewhere [38]. Briefly, an interview assessing cardiovascular risk factors was performed by telephone, followed by an in-person physical assessment 3–4 weeks after the telephone interview. The examination included blood pressure measurements, waist circumference measurements, BMI assessment, blood and urine samples and an electrocardiogram (ECG). The Institutional Review Boards of all participating institutions approved the study. There were 29,701 participants with baseline data and follow-up for all-cause and cancer-specific mortality outcomes included in the current analysis.

### 4.2. Exposure

Study participants were categorized as high risk or low risk for each biomarker, based on previously defined cut-off points [14] for the following biomarkers: serum albumin < 3.8 g/dL, C-reactive protein (CRP) > 3 mg/L, high-density lipoprotein (HDL) < 40 mg/dL, total cholesterol ≥ 240 mg/dL, heart rate ≥ 90 beats/min, systolic blood pressure ≥ 140 mmHg, diastolic blood pressure ≥ 90 mmHg, serum creatinine ≥ 1.3 mg/dL, and blood urea nitrogen (BUN) ≥ 18 mg/dL. Waist circumferences measured at the study visit (WC) > 88 cm in females and >102 cm in men, were utilized in place of waist-to-hip ratio, which was unavailable in the REGARDS dataset. AL score was determined by summing the number of high-risk biomarkers for each participant, and participants were considered to have high AL if the total score was ≥3. 

### 4.3. Outcome

The main outcomes of this study were all-cause mortality and cancer-specific mortality. Mortality was ascertained through bi-annual follow-up, linkage with the Social Security Death Index and the National Death Index, as well as medical records and death information from the participants’ proxies. Time to death was determined based on death certificates, the Social Security Death Index and the National Death Index, and final cause of death was defined after adjudication by the REGARDS clinical investigators using all available information [39]. Follow-up for this analysis was through to the end of the 2012 calendar year. 

### 4.4. Covariates

The self-reported study covariates included age (continuous), gender (male/female), race/ethnicity (Black/White), education (less than high school, high school, some college, college +), income (≥USD 75K, USD 35K–74K, USD 20K–34K, <USD 20K or refused), physical activity (none, 1–3 times/week, or ≥4 times/week), smoking (never, past, or current smoker) alcohol (none, moderate, or heavy intake), and self-reported type 2 diabetes (yes/no). BMI (weight (kg)/height (m)^2^) was calculated using height and weight measurements taken by an examiner at the patient’s in-home exam. Baseline comorbidity score was derived by summing the number of self-reported comorbid conditions present, including: hypertension, dyslipidemia, coronary artery disease, atrial fibrillation, myocardial infarction, diabetes, peripheral artery diseases, and stroke.

### 4.5. Statistical Analysis

Chi-square test, *t*-test, analysis of variance (ANOVA), or Kruskal–Wallis test were used to compare the distribution (mean and standard deviation (SD)) of baseline characteristics, and to describe the distribution (proportions, mean and SD, or median and interquartile range) of baseline participants’ characteristics by AL categories. Multiple imputation techniques were used to impute missing values in serum albumin and blood urea nitrogen. Missingness was determined based on albumin, BUN, systolic blood pressure, diastolic blood pressure, and heart rate. Based on the missing rate of approximately 30%, 25 imputations were determined to be sufficient to generate the final AL values for the analytic cohort without substantial loss of power or reliability in estimates [40]. After imputation, the SAS MIANALYZE procedure was used in combination with ANOVA, generalized linear modeling, quantile regression, and Cochran–Mantel–Haenszel testing to calculate pooled mean and median AL score by each defined category, and to compare group differences in the continuous AL score distributions pooled across imputations. A single imputation run was used to present cohort demographics and compare group differences by dichotomized high (≥3) vs. low AL score. Cox proportional hazard regression with a robust sandwich standard error estimator was used to estimate the hazard ratios (HR) and 95% confidence intervals (95% CIs) in each imputed dataset, and pooled HRs and 95% CIs were generated using the MIANALYZE procedure in SAS. Allostatic load score was modeled as a continuous variable in the main analyses. The crude model included allostatic load score only: model 1 adjusted for age; model 2 further adjusted for race; model 3 further adjusted for sex, education, and income; model 4 further adjusted for physical activity, smoking status, and alcohol consumption. Multivariable-adjusted survival models with interaction terms for allostatic load score and race, allostatic load score and BMI category, and allostatic load score and age group (<65 versus 65+) were run to detect the potential effect modification by these factors on the associations between allostatic load and mortality. The individual components of the AL score were modeled as continuous, or high vs. low, based on the AL definition criteria. Stratified analysis by BMI, race, and age group (<65 years and ≥65 years) were also conducted. *p*-values of ≤0.05 were considered statistically significant. All statistical analyses were conducted in SAS 9.4 (SAS Institute Inc., Cary, NC, USA). 

## 5. Conclusions

Blacks had higher baseline AL scores on average than Whites in the REGARDS cohort; higher AL score at baseline was associated with increased risk of all-cause and cancer-specific mortality among both Blacks and Whites. A significant association between AL score and all-cause mortality was observed, regardless of BMI and age-group. Given that AL reflects the whole body dysregulation of cardiovascular, immune, and metabolic systems, strategies designed to reduce psychosocial stress and associated behavioral risk factors may be useful as part of comprehensive mortality reduction strategies. 

## Figures and Tables

**Figure 1 cancers-12-01695-f001:**
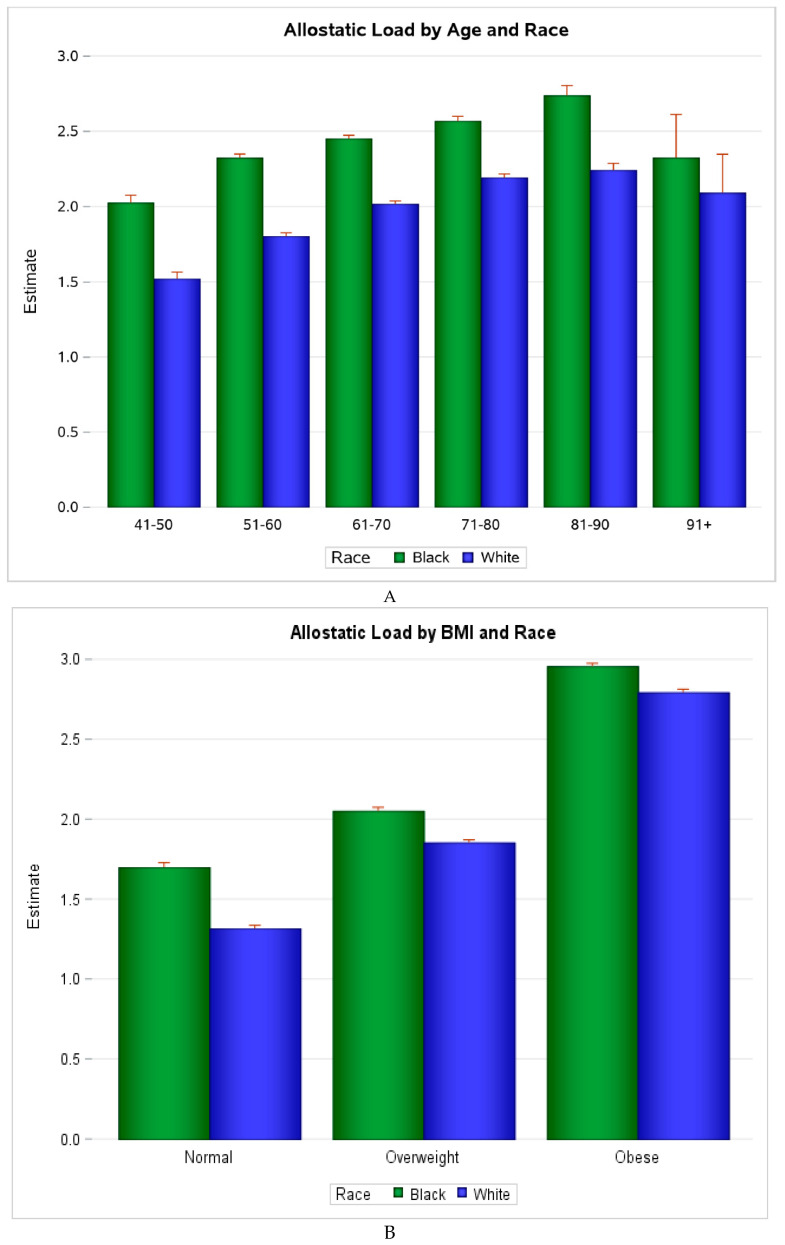
(**A**) Distribution of Allostatic Load Score in the REGARDS Cohort by Age and Race. (**B**). Distribution of Allostatic Load Score in the REGARDS Cohort by BMI and Race.3. Discussion

**Table 1 cancers-12-01695-t001:** Study characteristics ^§^ and Mean Allostatic Load Score ^‡^ stratified by Race in the REGARDS cohort (*N* = 29,701).

Characteristics ^§^		Black ^‡^	White ^‡^	*p*
Age, years, mean (SD)		64.1 (9.2)	65.4 (9.5)	<0.001
Comorbidity score, mean (SD)		2.02(1.36)	1.79(1.4)	<0.001
Allostatic load, mean (SD) *		2.42 (1.50)	1.99 (1.39)	<0.001
High allostatic load, % (SE) ^†^		44.6 (0.4)	33.4 (0.3)	<0.001
Allostatic load, mean (SD) by patient characteristics				
Sex	Female	2.49 (1.46)	1.88 (1.35)	<0.001
	Male	2.29 (1.56)	2.09 (1.42)	
Region	Stroke Belt	2.44 (1.49)	2.01 (1.40)	0.028
	Stroke Buckle	2.34 (1.48)	1.99 (1.37)	
	Non-Belt	2.43 (1.51)	1.97 (1.38)	
Annual Income	≥USD 75,000	1.98 (1.42)	1.63 (1.30)	<0.001
	USD 35,000 to USD 74,000	2.18 (1.45)	1.95 (1.36)	
	USD 20,000 to USD 34,000	2.47 (1.49)	2.21 (1.41)	
	<USD 20,000	2.68 (1.51)	2.38 (1.43)	
	Refused	2.52 (1.52)	1.89 (1.37)	
Education	College +	2.18 (1.48)	1.78 (1.34)	0.035
	Some college	2.35 (1.45)	2.05 (1.39)	
	High school	2.47 (1.51)	2.12 (1.39)	
	<High School	2.74 (1.51)	2.47 (1.44)	
Alcohol	None	2.50 (1.51)	2.14 (1.41)	0.724
	Moderate	2.20 (1.47)	1.81 (1.34)	
	Heavy	2.08 (1.33)	1.66 (1.32)	
Physical activity	None	2.68 (1.54)	2.28 (1.43)	0.19
	1–3 times/week	2.34 (1.46)	1.91 (1.36)	
	≥4 times /week	2.15 (1.44)	1.78 (1.33)	
Smoking	Never	2.39 (1.50)	1.87 (1.35)	0.087
	Past	2.46 (1.50)	2.08 (1.40)	
	Current	2.41 (1.49)	2.12 (1.43)	
Diabetes	Yes	2.94 (1.51)	2.79 (1.41)	<0.001
	No	2.20 (1.44)	1.84 (1.33)	
BMI	Normal	1.60 (1.35)	1.28 (1.13)	0.379
	Overweight/obese	2.60 (1.47)	2.28 (1.38)	
Biomarkers comprising AL score calculation				
BUN, mg/dL, median (IQR)		15.0 (12.0–19.0)	16.9 (13.6–20.0)	<0.001
Albumin, g/dL, mean (SD)		4.11(0.33)	4.21(0.32)	<0.001
CRP, mg/L, median (IQR)		3.1 (1.2–7.2)	1.9 (0.8–4.6)	<0.001
Creatinine, mg/dL, median (IQR)		0.9 (0.7–1.1)	0.8 (0.7–0.9)	<0.001
DBP, mmHg, mean (SD)		78.44(10.08)	75.16(9.19)	<0.001
SBP, mmHg, mean (SD)		130.77(17.35)	125.37(15.8)	<0.001
HDL, mg/dL, median (IQR)		51.0 (42.0–62.0)	48.0 (39.0–60.0)	<0.001

* Allostatic load score defined as sum score of the number of biomarkers above a set threshold including: serum albumin < 3.8 g/dL, C-reactive protein (CRP) > 3 mg/L, high-density lipoprotein (HDL) < 40 mg/dL, total cholesterol ≥ 240 mg/dL, heart rate ≥ 90 beats/min, systolic blood pressure ≥ 140 mmHg, diastolic blood pressure ≥ 90 mmHg, serum creatinine ≥ 1.3 mg/dL, and blood urea nitrogen (BUN) ≥ 18 mg/dL. Waist circumference (WC) > 88 cm in females and >102 cm in men. ^†^ High AL: ≥3 biomarkers above threshold for individual biomarkers. ^‡^ Presented mean (SD) or median (IQR) for continuous and count (percent) for categorical variables. *p*-values are from Kruskal–Wallis or ANOVA testing. ^§^ Demographic/behavioral variables and diabetes status are self-reported. Abbreviations: BMI, body mass index (kg/m^2^); BUN, blood urea nitrogen; CRP, C-reactive protein; DBP, diastolic blood pressure; HDL, high density lipoprotein; IQR, inter quartile range; SBP, systolic blood pressure; SD, standard deviation.

**Table 2 cancers-12-01695-t002:** Study Characteristics‡ by Allostatic Load Categories * in the REGARDS cohort (*N* = 29,701).

Characteristics‡		High AL (≥3) * (*N* = 11,280)	Low AL (<3) (*N* = 18,421)	*p*
Age, years, mean (SD)		66.0 (9.3)	64.2 (9.4)	**<0.001**
Sex (% female)		6274 (55.6)	10,088(54.7)	0.149
Race (% Black)		5454 (48.3)	6770 (36.7)	**<0.001**
Region	Stroke Belt	3980 (35.2)	6308 (34.2)	**0.009**
	Stroke Buckle	2261 (20.0)	3956 (21.4)	
	Non-Belt	5039 (44.6)	8157 (44.2)	
Annual Income	≥USD 75,000	1161 (10.3)	3537 (19.23)	**<0.001**
	USD 35,000 to USD 74,000	3040 (26.9)	5772 (31.3)	
	USD 20,000 to USD 34,000	3054 (27.1)	4123 (22.4)	
	<USD 20,000	2631 (23.3)	2715 (14.7)	
	Refused	1394 (12.4)	2274 (12.3)	
Education	College +	3180 (28.2)	7165 (38.9)	**<0.001**
	Some college	3012 (26.7)	4965 (26.9)	
	High school	3175 (28.1)	4494 (24.4)	
	<High School	1913 (16.9)	1797 (9.7)	
Alcohol	None	7927 (70.3)	10,902 (59.2)	**<0.001**
	Moderate	3022 (26.8)	6673 (36.2)	
	Heavy	331 (2.9)	846 (4.6)	
Physical activity	None	4685 (41.5)	5385 (29.2)	**<0.001**
	1–3 times/week	3900 (34.6)	7079 (38.4)	
	≥4 times/week	2695 (23.9)	5957 (32.3)	
Smoking	Never	4811 (42.6)	8690 (47.2)	**<0.001**
	Past	4689 (41.6)	7226 (39.2)	
	Current	1780 (15.8)	2505 (13.6)	
Diabetes (% yes)		3630 (32.2)	2664 (14.5)	**<0.001**
BMI	Normal	1363 (12.1)	6150 (33.4)	**<0.001**
	Overweight/obese	9917 (87.9)	12,271 (66.6)	
BUN, mg/dL, median (IQR)		19.0 (14.6–23.0)	15.0 (12.0–18.0)	**<0.001**
Comorbid score, mean (SD)		2.5 (1.4)	1.5 (1.3)	**<0.001**
Albumin, g/dL, mean (SD)		4.1 (0.4)	4.2 (0.3)	**<0.001**
CRP, mg/L, median (IQR)		4.8 (2.5–9.0)	1.6 (0.8–3.2)	**<0.001**
Creatinine, mg/dL, median (IQR)		0.9 (0.8–1.2)	0.8 (0.7–1.0)	**<0.001**
DBP, mmHg, mean (SD)		79.5 (11.0)	74.7 (8.4)	**<0.001**
SBP, mmHg, mean (SD)		135.1 (18.3)	123.0 (13.7)	**<0.001**
HDL, mg/dL, median (IQR)		44.0 (36.0–56.0)	52.0 (43.9–64.0)	**<0.001**

* High AL: ≥3 biomarkers above threshold for individual biomarkers, † Demographic/behavioral variables and diabetes status are self-reported, ‡ *p*-values are from chi-squared test, *t*-test, or Kruskal–Wallis test, Bold text *p*-values are below the selected two-sided alpha of 0.05.

**Table 3 cancers-12-01695-t003:** Multivariable Adjusted Cox Proportional Hazards Models * for the Association ^†‡Ω^ of Allostatic Load Score with All-Cause and Cancer-Specific Mortality, Overall and by Race in the REGARDS cohort.

Models	Overall (*N* = 29,701) ^†‡^		Black (*N* = 12,224) ^†‡^		White (*N* = 17,476) ^†‡^	
Allostatic Load	All-Cause Mortality ^‡^	Cancer mortality ^‡^	All-Cause mortality ^‡^	Cancer Mortality ^‡^	All-Cause mortality ^‡^	Cancer mortality ^‡^
Deaths	4622	1237	2055	513	2607	724
Crude *	**1.35 (1.33, 1.38)**	**1.15 (1.11, 1.19)**	**1.32 (1.28, 1.36)**	**1.09 (1.03, 1.16)**	**1.37 (1.34, 1.41)**	**1.20 (1.14, 1.26)**
Model 1 *	**1.32 (1.29, 1.34)**	**1.11 (1.07, 1.16)**	**1.29 (1.25, 1.33)**	1.06 (1.00, 1.13)	**1.31 (1.28, 1.35)**	**1.14 (1.09, 1.21)**
Model 2 *	**1.30 (1.28, 1.33)**	**1.11 (1.06, 1.15)**	-	-	-	-
Model 3 *	**1.27 (1.24, 1.30)**	**1.08 (1.04, 1.13)**	**1.33 (1.29, 1.37)**	**1.11 (1.04, 1.18)**	**1.31 (1.28, 1.35)**	**1.15 (1.09, 1.22)**
Model 4 *	**1.24 (1.22,1.27)**	**1.07 (1.03,1.12)**	**1.26 (1.22,1.30)**	**1.06 (0.99,1.13)**	**1.23 (1.20,1.27)**	**1.08 (1.03,1.14)**
Race ^Ω^						
Black	1.05 (0.99, 1.12)	1.03 (0.92,1.17)	-		-	-
White (Ref)			-		-	-

*All models included allostatic load score as the main covariate. Model 1 additionally adjusted for age at baseline, model 2 further adjusted for race, model 3 further adjusted for sex, education, and income, and model 4 further adjusted for smoking status, physical activity level, and alcohol consumption. Models stratified by race include all covariables as described with race excluded. ^†^ Analysis was conducted overall for all participants and stratified by race. ^‡^ Results presented as HRs and 95% CI. ^Ω^ Presented patient covariate HRs and 95% are from fully adjusted multivariable model (model 4). Bold text text *p*-values are below the selected two-sided alpha of 0.05.

**Table 4 cancers-12-01695-t004:** Multivariable Adjusted Cox Proportional Hazards Models * for the Association ^†‡^ of Allostatic Load Score with All-Cause and Cancer-Specific Mortality in REGARDS Stratified by BMI (*N* = 29,701).

Allostatic Load	Normal BMI (*n* = 7299) ^†^		Overweight/Obese (*N* = 22,401) ^†^	
	All-Cause mortality ^‡^	Cancer mortality ^‡^	All-Cause mortality ^‡^	Cancer mortality ^‡^
Deaths	1414	407	3248	830
Crude *	**1.56 (1.50, 1.63)**	**1.40 (1.30, 1.50)**	**1.41 (1.38, 1.44)**	**1.17 (1.11, 1.23)**
Model 1 *	**1.41 (1.35, 1.47)**	**1.29 (1.19, 1.39)**	**1.36 (1.33, 1.40)**	**1.13 (1.07, 1.18)**
Model 2 *	**1.38 (1.32, 1.44)**	**1.27 (1.17, 1.37)**	**1.35 (1.32, 1.39)**	**1.12 (1.07, 1.18)**
Model 3 *	**1.34 (1.28, 1.40)**	**1.22 (1.13, 1.33)**	**1.32 (1.29, 1.35)**	**1.10 (1.05, 1.16)**
Model 4 *	**1.29 (1.23, 1.35)**	**1.17 (1.08, 1.28)**	**1.29 (1.26, 1.32)**	**1.09 (1.03, 1.14)**
Race ^Ω^				
Black	**1.18 (1.05, 1.33)**	1.11 (0.89, 1.38)	**1.09 (1.01, 1.17)**	1.06 (0.91, 1.23)
White (Ref)	-	-	-	-

* All models included allostatic load score as main covariate. Model 1 additionally adjusted for age at baseline, model 2 further adjusted for race, model 3 further adjusted for sex, education, and income, and model 4 further adjusted for smoking status, physical activity, and alcohol consumption. ^†^ Analysis was stratified by BMI. ^‡^ Results presented as HRs and 95% C. ^Ω^ Presented patient covariate HRs and 95% are from fully adjusted multivariable model (model 4). Bold text *p*-values are below the selected two-sided alpha of 0.05.

**Table 5 cancers-12-01695-t005:** Multivariable Adjusted Cox Proportional Hazards Models * for the Association ^†‡^ of Allostatic Load Score with All-Cause and Cancer-Specific Mortality in REGARDS Stratified by Age Group (*N* = 29,700).

Allostatic Load	Age < 65 years (*N* = 15,008) ^†^		Age ≥65 years (*N* = 14,692) ^†^	
	All-Cause mortality ^‡^	Cancer mortality ^‡^	All-Cause mortality ^‡^	Cancer mortality ^‡^
Deaths	1106	320	3556	917
Crude	**1.47 (1.42, 1.53)**	**1.17 (1.09, 1.26)**	**1.28 (1.25, 1.31)**	**1.11 (1.06, 1.16)**
Model 1 *	**1.46 (1.40, 1.51)**	**1.15 (1.07, 1.24)**	**1.27 (1.24, 1.30)**	**1.10 (1.05, 1.15)**
Model 2 *	**1.43 (1.37, 1.49)**	**1.13 (1.04, 1.22)**	**1.26 (1.23, 1.29)**	**1.10 (1.05, 1.15)**
Model 3 *	**1.36 (1.31,1.42)**	**1.07(0.99, 1.16)**	**1.24 (1.21, 1.27)**	**1.08 (1.03, 1.13)**
Model 4 *	**1.34 (1.28, 1.39)**	**1.06 (0.98, 1.15)**	**1.22 (1.19, 1.25)**	**1.07 (1.02, 1.12)**

* All models included allostatic load score as main covariate. Model 1 additionally adjusted for age at baseline, model 2 further adjusted for race, model 3 further adjusted for sex, education, and income. ^†^ Analysis was stratified by age < 65 years and ≥65 years. ^‡^ Results presented as HRs and 95% CI. Bold text *p*-values are below the selected two-sided alpha of 0.05.

**Table 6 cancers-12-01695-t006:** Multivariable Adjusted Cox Proportional Hazards Models *^‡^ for AL Components with All-Cause and Cancer-Specific Mortality (*N* = 29,701).

Allostatic Load Components	Continuous *	High vs. Low *^†^	Continuous *	High vs. Low *^†^
	All-Cause mortality ‡		Cancer mortality ‡	
**Overall**				
Deaths	4622	4622	1237	1237
CRP	**1.01 (1.01, 1.02)**	**1.50 (1.41, 1.59)**	**1.01 (1.01, 1.02)**	**1.30 (1.16, 1.46)**
Albumin	**0.60 (0.53, 0.67)**	**1.44 (1.31, 1.59)**	0.80 (0.64, 1.00)	1.12 (0.90, 1.39)
BUN	**1.03 (1.02, 1.04)**	**1.46 (1.35, 1.57)**	1.01 (1.00, 1.02)	1.12 (0.97, 1.28)
Total Cholesterol	1.00 (1.00,1.00)	1.00 (0.90, 1.11)	1.00 (1.00, 1.00)	0.95 (0.78, 1.16)
Creatinine	**1.37 (1.30, 1.43)**	**2.10 (1.93, 2.28)**	**1.14 (1.03, 1.25)**	1.19 (0.98, 1.45)
HDL	1.00 (0.99, 1.00)	**1.13 (1.05, 1.21)**	1.00 (0.99, 1.00)	1.07 (0.93, 1.23)
DBP	1.00 (1.00, 1.00)	**1.15 (1.03, 1.28)**	0.99 (0.99, 1.00)	0.82 (0.65, 1.03)
SBP	1.01 (1.00,1.01)	**1.29 (1.21, 1.38)**	1.00 (1.00, 1.00)	1.02 (0.90, 1.17)
WC	**1.01 (1.01, 1.01)**	1.06(0.99,1.14)	1.00 (1.00, 1.00)	0.91 (0.80, 1.04)
Blacks				
Deaths	2055	2055	513	513
CRP	**1.01 (1.01, 1.02)**	**1.37 (1.24, 1.50)**	**1.01 (1.01, 1.02)**	1.08 (0.90, 1.29)
Albumin	**0.52 (0.44, 0.61)**	**1.57 (1.38, 1.78)**	0.74 (0.52, 1.05)	1.12 (0.83, 1.50)
BUN	**1.03 (1.02, 1.04)**	**1.57 (1.42, 1.75)**	1.01 (1.00, 1.02)	1.13 (0.92, 1.39)
Total Cholesterol	1.00 (1.00, 1.00)	1.06 (0.92, 1.23)	1.00 (1.00, 1.00)	0.98 (0.73, 1.31)
Creatinine	**1.34 (1.27, 1.41)**	**2.25 (2.00, 2.52)**	**1.15 (1.04, 1.27)**	1.24 (0.95, 1.62)
HDL	1.00 (0.99, 1.00)	**1.15 (1.03, 1.29)**	1.00 (0.99, 1.00)	1.12 (0.89, 1.40)
DBP	1.00 (1.00, 1.01)	**1.20 (1.05, 1.37)**	0.99 (0.98, 1.00)	0.88 (0.65, 1.17)
SBP	**1.01 (1.01, 1.01)**	**1.34 (1.22, 1.47)**	1.00 (1.00, 1.01)	1.08 (0.89, 1.31)
WC	1.01 (1.01, 1.01)	0.98 (0.88, 1.08)	1.00 (0.99, 1.00)	0.89(0.73, 1.09)
Whites				
Deaths	2607	2607	704	704
CRP	**1.02 (1.02, 1.03)**	**1.61 (1.48, 1.74)**	**1.02 (1.01, 1.02)**	**1.48 (1.27, 1.72)**
Albumin	**0.68 (0.58, 0.80)**	**1.32 (1.15, 1.53)**	0.84 (0.63, 1.13)	1.15 (0.86, 1.54)
BUN	**1.03 (1.03, 1.04)**	**1.37 (1.23, 1.52)**	1.00 (0.99,1.02)	1.11 (0.93, 1.33)
Total Cholesterol	1.00 (1.00, 1.00)	0.95 (0.82, 1.09)	1.00 (1.00, 1.00)	0.93 (0.71, 1.21)
Creatinine	**1.45 (1.34, 1.58)**	**1.93 (1.70, 2.19)**	1.08 (0.86, 1.36)	1.11 (0.83, 1.48)
HDL	1.00 (0.99, 1.00)	**1.11 (1.01, 1.21)**	1.00 (0.99, 1.00)	1.04 (0.88, 1.24)
DBP	1.00 (0.99, 1.00)	1.06 (0.88, 1.27)	0.99 (0.98, 1.00)	0.74 (0.51, 1.08)
SBP	1.00 (1.00, 1.01)	**1.25 (1.14, 1.37)**	1.00 (0.99, 1.00)	0.98 (0.81, 1.17)
WC	**1.01 (1.01, 1.01)**	**1.14 (1.04, 1.26)**	1.00 (1.00, 1.01)	0.94 (0.79, 1.13)

* Model adjusted for demographic covariates, including age, race (except in race stratified models), sex, education, income, smoking status, physical activity level, and alcohol consumption,.† High AL: ≥3 biomarkers above threshold. Low AL: <3 biomarkers above threshold. ‡ Results are presented as HRs and 95% CI. Bold text *p*-values are below the selected two-sided alpha of 0.05.

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
