# Peer review of "Association of Allostatic Load with All-Cause and Cancer Mortality by Race and Body Mass Index in the REGARDS Cohort"

_cancers, 2020, doi:10.3390/cancers12061695_

Round 1
Reviewer 1 Report
Thank you to the study authors for responding to my initial comments. My one suggestion for further improvement is to be sure that the interpretation/conclusions are consistent throughout the paper. The results state that "there was evidence of effect modification by BMI and age-group," but the conclusion states that "A significant association between AL score and all-cause mortality was observed regardless of BMI and age-group." These statements seem to be in conflict. It appears to me that there may be some slight effect modification by BMI for cancer mortality, and slight effect modification by age for all-cause mortality, but AL increases risk of these outcomes across strata, so the associations are not qualitatively different. I suggest modifying the results section to better match the conclusion, which is clear and well-stated.
Author Response
Thank you for your feedback! We have revised our discussion of the effect modification in the results sections and added the following sentence to address this: "While magnitudes of the effect of AL score on mortality differed slightly across BMI and age strata, AL score consistently increased risk of mortality for all examined patient groups."
Reviewer 2 Report
The authors have made positive changes in response to reviewer comments, and the manuscript is much improved. Two (very) minor comments:
- Regarding Reviewer 1’s comment about examining cancer-specific deaths. I agree with the authors that this might be an excellent topic for future research. Linkage with local cancer registries individually, or through the North American Association of Central Cancer Registries’ budding Virtual Pooled Registry system might prove fruitful for this endeavor.
- In tables 3 and 4, are the stratified analyses presented by race (black, versus white reference) also multi-variable adjusted? This was not clear to me. Additional text in the footnote would be helpful.
Author Response
- Thank you for these suggestions! We will definitely look into these resources for our future work building on these findings.
- These estimates are also multivariable adjusted. We have edited the footnotes for these tables to clarify this.